# Polycystic Ovary Syndrome and Oxidative Stress—From Bench to Bedside

**DOI:** 10.3390/ijms241814126

**Published:** 2023-09-15

**Authors:** Natalia Zeber-Lubecka, Michał Ciebiera, Ewa E. Hennig

**Affiliations:** 1Department of Gastroenterology, Hepatology and Clinical Oncology, Centre of Postgraduate Medical Education, 01-813 Warsaw, Poland; ehennig@cmkp.edu.pl; 2Department of Genetics, Maria Sklodowska-Curie National Research Institute of Oncology, 02-781 Warsaw, Poland; 3Second Department of Obstetrics and Gynecology, Centre of Postgraduate Medical Education, 01-813 Warsaw, Poland; michal.ciebiera@cmkp.edu.pl; 4Warsaw Institute of Women’s Health, 00-189 Warsaw, Poland

**Keywords:** oxidative stress, polycystic ovary syndrome, oxidative phosphorylation, reactive oxygen species

## Abstract

Oxidative stress (OS) is a condition that occurs as a result of an imbalance between the production of reactive oxygen species (ROS) and the body’s ability to detoxify and neutralize them. It can play a role in a variety of reproductive system conditions, including polycystic ovary syndrome (PCOS), endometriosis, preeclampsia, and infertility. In this review, we briefly discuss the links between oxidative stress and PCOS. Mitochondrial mutations may lead to impaired oxidative phosphorylation (OXPHOS), decreased adenosine triphosphate (ATP) production, and an increased production of ROS. These functional consequences may contribute to the metabolic and hormonal dysregulation observed in PCOS. Studies have shown that OS negatively affects ovarian follicles and disrupts normal follicular development and maturation. Excessive ROS may damage oocytes and granulosa cells within the follicles, impairing their quality and compromising fertility. Impaired OXPHOS and mitochondrial dysfunction may contribute to insulin resistance (IR) by disrupting insulin signaling pathways and impairing glucose metabolism. Due to dysfunctional OXPHOS, reduced ATP production, may hinder insulin-stimulated glucose uptake, leading to IR. Hyperandrogenism promotes inflammation and IR, both of which can increase the production of ROS and lead to OS. A detrimental feedback loop ensues as IR escalates, causing elevated insulin levels that exacerbate OS. Exploring the relations between OS and PCOS is crucial to fully understand the role of OS in the pathophysiology of PCOS and to develop effective treatment strategies to improve the quality of life of women affected by this condition. The role of antioxidants as potential therapies is also discussed.

## 1. Introduction

Oxidative stress (OS), or an imbalance between oxidants and antioxidants, is related to the generation of excessive amounts of reactive oxygen species (ROS) and the body’s ability to defend against their damaging effects with antioxidants, which cause DNA damage and/or cell apoptosis, affect gene expression and the immune response [1]. Importantly, OS plays a key role in the pathophysiology of a variety of gynecological disorders, including polycystic ovary syndrome (PCOS), endometriosis, unexplained infertility, and preeclampsia [2,3].

PCOS is a common hormonal disorder, according to World Health Organization (WHO) data—with the prevalence of between 8% and 13% depending on the population studied [4]. It affects mostly women of reproductive age. It is characterized by a combination of symptoms associated with hormonal imbalances, menstrual irregularities, and the presence of excessive amounts of follicles in ovaries. The exact cause of PCOS is not fully understood, but it is believed to encompass a combination of genetic and environmental factors [5].

The diagnosis of PCOS is based on specific criteria established by the Rotterdam criteria or the Androgen Excess and PCOS Society (AE-PCOS Society) criteria [6]. For a diagnosis of PCOS according to the Rotterdam criteria, at least two out of three of the following must be met: 1. The presence of irregular menstrual cycles, which can include oligomenorrhea (infrequent periods) or amenorrhea (absence of periods); 2. Clinical signs of androgen excess, such as hirsutism, acne, or male-pattern baldness and/or laboratory evidence of elevated androgen levels; and 3. The presence of multiple small cysts in the ovaries, as visualized on ultrasound examination [6].

PCOS may occur in four different phenotypes or subtypes defined by specific clinical features and hormonal profiles [7]. Classic phenotype is characterized by the presence of both hyperandrogenism and chronic anovulation. Women with classic phenotype typically exhibit symptoms connected with excessive androgen levels such as hirsutism (excessive hair growth), acne, and menstrual irregularities. In contrast, women with ovulatory phenotype exhibit normal ovulatory function and no chronic anovulation. However, they may still present symptoms of hyperandrogenism, such as hirsutism or acne. This phenotype is diagnosed based on the presence of hyperandrogenism in the absence of anovulation [8]. Non-hyperandrogenic phenotype is characterized by the occurrence of chronic anovulation without significant signs of hyperandrogenism [9]. The last, non-obese phenotype is found in women who develop PCOS but do not meet the criteria for obesity [9].

Treatment options for PCOS may include lifestyle modifications such as weight loss, exercise, and a healthy diet, medications including hormonal contraceptives with anti-androgen activities, metformin, and anti-androgens, and fertility treatment for women who wish to conceive. Managing PCOS may help improve overall health outcomes and prevent long-term complications associated with the disorder, especially in older ages [10]. The current review summarizes some links between OS and PCOS.

## 2. ROS Production

Mitochondria, the energy-producing organelles, are a major source of ROS. In PCOS, mitochondrial dysfunction was observed, leading to an increased ROS production.

ROS play a dual role, acting as both beneficial signaling molecules and potentially harmful oxidative agents when their levels become excessive. ROS are primarily generated as byproducts of cellular respiration, particularly during the electron transport chain in mitochondria. This is a natural and essential process for energy production in cells. ROS encompass a range of molecules, including superoxide anion (O_2_^−^), hydroxyl radical (OH^−^), singlet oxygen, hydrogen peroxide (H_2_O_2_), organic hydroperoxide (ROOH), alkoxy and peroxy radicals (RO and ROO), hypochlorous acid (HOCl), and peroxynitrite (ONOO) [11,12]. In a healthy state, the production of ROS is balanced by the body’s antioxidant defense systems. These antioxidants include enzymes such as superoxide dismutase (SOD), catalase (CAT), and glutathione peroxidase (GPx), as well as non-enzymatic molecules such as glutathione and vitamins C and E [13,14]. This balance is crucial. ROS are involved in various aspects of ovarian physiology, serving as secondary messengers in cellular signaling pathways.

ROS play a nuanced role in ovarian physiology, where controlled levels of ROS are essential for normal ovarian function. However, an imbalance in ROS production and antioxidant defenses can lead to OS, which may have detrimental effects on reproductive outcomes. They play roles in regulating key ovarian processes, including meiosis, which is essential for the development of mature oocytes; ovulation, where ROS participate in the signaling cascades that trigger the release of the mature egg; and corpus luteum maintenance and regression [15].

The specific mechanism of ROS production in PCOS is not fully understood, but several factors contribute to increased ROS production in individuals with PCOS. When mitochondria are not functioning optimally, they generate more ROS as a byproduct of the electron transport chain during oxidative phosphorylation (OXPHOS). This excess ROS production contribute to OS in ovarian tissues [16]. In addition, numerous individuals with PCOS develops insulin resistance (IR). In patients with IR, hyperglycemia leads to an upsurge in the production of ROS through a NADPH oxidase p47(phox) component [17]. Furthermore, hyperglycemia promotes the release of tumor necrosis factor-alpha (TNF-α), a well-known contributor to IR, from mononuclear cells. It also enhances the activity of nuclear factor kappa-light-chain-enhancer of activated B cells (NF-κB) [18]. NF-κB, in turn, exacerbates OS by stimulating NADPH oxidase, which amplifies ROS production and sustains the inflammatory response [19]. Hyperandrogenemia heightens the sensitivity of white blood cells to hyperglycemia and aggravate OS [20]. Additionally, NADPH oxidase plays a critical role in generating ROS in individuals with obesity. In adipocytes, increased levels of fatty acids trigger OS by activating NADPH oxidase [21].

OS itself perpetuate further ROS production. Damage cellular components, including DNA, proteins, and lipids resulting ROS production, trigger cellular responses that generate more ROS as a protective mechanism, creating a feedback loop of OS [22,23]. It has also been noted that some women with PCOS have reduced antioxidant defense mechanisms, which can lead to an imbalance between ROS production and antioxidant defense, resulting in increased OS [24,25]. In addition, an imbalance in luteinizing hormone (LH), follicle-stimulating hormone (FSH), and other reproductive hormones can affect ovarian function and may contribute to ROS production [26].

Lipid peroxidation (LPO) is a biochemical process in which reactive substances such as free radical’s target lipids, particularly those containing carbon–carbon double bonds, notably polyunsaturated fatty acids (PUFAs) [27]. Damage to cellular structures, including cell membranes, as a result of PUFA-induced OS, leads to the accumulation of LPO products such as malondialdehyde (MDA) and the hydroxyl radical. This heightened OS often results in increased levels of ROS in the bloodstream [11]. In PCOS, alterations in the LPO processes appear to be compensatory mechanisms, which is evident through an increase in the concentrations of antioxidants such as α-tocopherol and retinol [28]. Additionally, there is a modest reduction in the activity of SOD enzyme responsible for combating OS. These adaptations may serve to counteract the elevated OS associated with PCOS and help maintain cellular integrity and function [28,29]. LPO potentially affect the quality of oocytes. High levels of OS and LPO may lead to DNA damage in oocytes, which can reduce fertility and increase the risk of miscarriage [30].

## 3. Mitochondrial DNA Damage

Elevated ROS may cause damage to mitochondrial DNA (mtDNA) and impair cellular energy production, contributing to metabolic and hormonal abnormalities seen in PCOS. Mitochondria are controlled both by mitochondrial and nuclear genomes. MtDNA is circular and double-stranded. It comprises a molecule which is 16,569 bp in length and encodes 22 tRNAs, 2 rRNAs and 13 polypeptides that are essential for adenosine triphosphate (ATP) production [31,32]. Unlike nuclear DNA, which is located in the cell nucleus and inherited from both parents, mtDNA is inherited in a maternal pattern only [33]. Due to its unique inheritance pattern, mtDNA may be used to trace maternal lineages and study human population genetics. It is less prone to recombination and has a higher mutation rate compared to nuclear DNA, making it useful for analyzing evolutionary relationships and population history [31,34]. Any defects at the level of mtDNA replication affect the formation of numerous mutations, whereas mtDNA exhibits a lack of histone protection and a DNA damage repair system [35]. Indeed, previous studies showed that mtDNA mutations contributed to numerous diseases, including PCOS [36] (Figure 1A).

The following seven variants in genes encoding mt-tRNA genes were identified in women with PCOS: tRNAGln (*MT-TQ*), tRNACys (*MT-TC*), tRNAAsp (*MT-TD*), tRNALys (*MT-TK*), tRNAArg (*MT-TR*), tRNAGlu (*MT-TE*), and tRNASer (UCN) (*MT-TS1*), in addition to seven variants in the 12S rRNA and three variants in 16S rRNA genes [37,38]. It is suggested that mutations affecting highly conserved nucleotides such as found in the mt-tRNA or OXPHOS complex genes, critical for their stability and biochemical function, may lead to mitochondrial dysfunction and are likely to be involved in the pathogenesis of PCOS [37,38].

For years, a team of Chinese researchers looking for mutations in mtDNA had repeatedly shown that mitochondrial dysfunction caused by mt-tRNA mutations might be involved in the pathogenesis of PCOS-IR [39,40]. The nine mt-tRNA mutations were shown that were potentially associated with PCOS-IR: tRNALeu (UUR) (*MT-TL1*) A3302G and C3275A mutations, tRNAGln (*MT-TQ*) T4363C and T4395C mutations, tRNASer (UCN) (*MT-TS1*) C7492T mutation, tRNAAsp (*MT-TD*) A7543G mutation, tRNALys (*MT-TK*) A8343G mutation, tRNAArg (*MT-TR*) T10454C mutation and tRNAGlu (*MT-TE*) A14693G mutation.

A south Indian female population showed a significant association between D310 and A189G variants in the mtDNA D-loop region, which is much more variable and non-coding, and the reduction in mtDNA copy number (mtCN) in PCOS group compared to the controls [41]. Conversely, in the Korean population mtCN was significantly lower in women with PCOS compared to the controls. The correlation was negative for IR and positive for sex hormone–binding globulin (SHBG) levels [42].

It is important to recognize that mitochondrial mutations are just one aspect of mitochondrial dysfunction and are not widely recognized as the primary cause of PCOS. The majority of PCOS cases are considered to involve complex genetic, hormonal, and environmental factors [43].

## 4. Oxidative Stress and PCOS

The first studies demonstrating elevated levels of OS and reduced antioxidant capacity in women with PCOS were published more than 20 years ago [44]. The finding was particularly relevant in women with obesity-associated PCOS phenotype, hyperandrogenism or the development of metabolic syndrome [45,46,47]. In addition, an in vitro study showed that OS increased the activity of ovarian steroid-producing enzymes and stimulated androgen production [48].

The process of ovulation involves the rupture and release of the dominant follicle from the ovary into the fallopian tube, where fertilization may occur [49]. The regulation of ovulation is influenced by the fluctuating levels of gonadotropic hormones, especially FSH and LH, which are released by the pituitary gland [49].

The ovulation process is the key phase of the menstrual cycle [50]. It follows the follicular phase, during which the dominant follicle develops and matures under the influence of FSH. The dominant follicle then releases the egg in response to LH surge, marking the occurrence of ovulation. Following ovulation, the luteal phase begins, characterized by the formation of the corpus luteum from the remnants of the ruptured follicle. The corpus luteum produces progesterone, which helps prepare the uterine lining for a potential implantation of a fertilized egg. If fertilization and implantation do not occur, the corpus luteum degenerates, leading to a decline in progesterone levels [51]. This hormonal shift triggers the shedding of the uterine lining, resulting in menstruation. The timing of ovulation within the menstrual cycle is typically around 14 days before the start of menstruation in a regular, 28-day cycle. However, it is important to note that the duration of the menstrual cycle and the timing of ovulation may vary among individuals. Factors such as stress, hormonal imbalances, and certain medical conditions, including PCOS, may disrupt the normal regulation of ovulation [51].

Some authors described the involvement of ROS in the ovulation process. Elevated secretion of LH in the lead-up to ovulation triggers the release of inflammatory substances within the ovary [52]. This, in turn, leads to an overproduction of ROS. Increased ROS levels play crucial roles in key aspects of the ovulation process, including cumulus expansion, progesterone production, expression of preovulatory genes and activation of ovulatory signals [53]. The interplay between ROS and SOD in the corpus luteum is pivotal in determining the duration and efficiency of progesterone production [54]. Enhanced SOD activity acts as a protector against ROS-induced damage, while reduced SOD activity can lead to ROS-triggered apoptosis and the regression of the corpus luteum. These dynamics highlight the central roles of ROS and SOD in governing ovarian physiology and reproductive health [54]. The administration of broad-range scavengers of oxidative species into the ovarian bursa of mice which were hormonally induced to ovulate resulted in a significant reduction in the rate of ovulation [55]. The study also revealed that antioxidants prevented LH-induced cumulus expansion, which is a necessary requirement for ovulation. Such an effect was observed both in vivo and, in an ex, vivo system using isolated intact ovarian follicles. Those findings suggested that OS played a role in the ovulation process. By administering antioxidants to neutralize those oxidative species, the rate of ovulation was reduced, and cumulus expansion was prevented [55].

Abnormal ovulation, associated with cell apoptosis, and reduced follicular atresia is a major aspect of PCOS. The abnormal control of caspase 9, which is a key player in the intrinsic or mitochondrial pathway and is involved in apoptosis, contributes to the elevation of total oxidant status. Studies showed that decreased levels of certain apoptosis markers and increased OS might be involved in the progression of PCOS [56]. In addition, studies indicated that an imbalance between antioxidant factors and ROS in the follicular fluid might affect oocyte quality, thereby resulting in abnormal ovulation and infertility in patients with PCOS [57,58].

Hyperandrogenism is a medical condition characterized by increased levels of androgens, which are male sex hormones but are also present in smaller amounts in females. These androgens include testosterone, dihydrotestosterone (DHT), androstenedione, dehydroepiandrosterone (DHEA), and dehydroepiandrosterone sulfate (DHEA-S). In women, androgens are produced by the ovaries, adrenal glands, skin, subcutaneous tissues, and liver [59]. The occurrence of hyperandrogenemia (HA) in women is relatively common, affecting about 5% to 10% of the female population [59]. The primary cause of HA is PCOS, which accounts for approximately 80% of cases [60]. Other causes of HA can include medications, hormone-producing tumors, or excessive natural production of androgens [7]. Hormonal imbalance can lead to a range of physical symptoms, including acne, hirsutism, alopecia, and menstrual irregularities and IR [7]. Treatment approaches aim to address the underlying cause and manage the specific symptoms experienced by the individual [5].

In women with PCOS, there is an excess of testosterone [61]. Despite the presence of higher levels of a precursor hormone—androstenedione (A4), testosterone remains the most abundant and potent conventional androgen circulating in PCOS women’s bodies. Traditionally, the ovaries have been recognized as the primary source of excessive conventional androgens in women with PCOS. However, some women with PCOS also experience an overproduction of androgens from their adrenal glands [62]. The adrenal cortex is responsible for generating not only conventional androgens but also a group of androgen metabolites, 11-oxygenated androgens [63]. This includes 11-ketotestosterone (11-ketoT), which is equally potent as testosterone and contributes to the overall androgen excess seen in PCOS [63]. Furthermore, in women with PCOS, as well as in hyperandrogenic or obese adolescent girls, there is an elevation in both circulating and peripheral tissue levels of 11-ketoT [64]. This increase is likely due to the higher expression of a specific enzyme, aldo-keto reductase family 1 member C3 (AKR1C3 or 17β-hydroxysteroid dehydrogenase type 5) [65]. Importantly, 11-ketoT is a nonaromatizable androgen, which means it cannot be converted into estrogen. It also has a low affinity for SHBG, making it more bioavailable in the body. Additionally, 11-ketoT can cross the placental barrier and affect the fetus [63].

Increased frequency of LH pulse secretion led to excessive androgen production in the theca-interstitial cells of the ovaries [66]. Elevated insulin levels in IR directly or in synergy with LH stimulate the growth of theca-interstitial cells in the ovaries and inhibit the production of SHBG, which binds to sex hormones in the bloodstream, making them less bioavailable [66]. High levels of androgens can promote inflammation and IR, both of which can increase the production of ROS and lead to OS. Increased OS and activate NF-κB levels result in a state of chronic low-grade inflammation, which promotes the expression of various inflammatory molecules, including TNF-α and interleukin-6 (IL-6) [67]. In ovaries, TNF-α, can stimulate the proliferation of theca-interstitial cells [68]. Additionally, in the inflamed state, there is an upregulation of cytochrome P450-17α-hydroxylase (CYP17), an enzyme involved in androgen production, leading to increased androgens within the ovaries [69]. In turn, in OS state, the expression of hepatocyte nuclear factor-4α (HNF-4α), regulator of androgen biology, is reduced, resulting in decreased levels of SHBG, which contribute to HA [70]. OS impair insulin signaling pathways within cells, exacerbating IR. This creates a vicious cycle where IR leads to higher insulin levels, further worsening OS. Elevated androgen levels contribute to IR by interfering with insulin signaling in cells. This makes it more challenging for glucose to enter cells, leading to higher blood sugar levels [71].

The oocyte obtains energy necessary to complete the maturation process from the cumulus-oocyte complex due to, among others, the citric acid (TCA) cycle [72]. The disruption of citrate formation in this process was shown to indicate reduced oocyte competence [73]. A study was conducted in a mouse model of hyperandrogenism induced by dietary supplementation with DHEA, an androgen precursor. It revealed that citrate levels, glucose-6-phosphate dehydrogenase activity and lipid content in the oocytes of those mice were significantly lower than in the controls without PCOS, implying the abnormal metabolism of TCA and the pentose phosphate pathway. This suggests that elevated DHEA levels may have a negative impact on oocyte function and contribute to impaired pregnancy in women with hyperandrogenism and PCOS [74] (Figure 1B). Another study based on hepatic exosome metabolome, abnormalities within which were linked to the pathophysiology of PCOS [75,76], showed that reduced levels of glycolysis and TCA cycle were observed in a mouse model of PCOS regardless of age [77].

## 5. Oxidative Phosphorylation and PCOS

Mitochondria belong to the key players in generating and regulating cellular bioenergetics, producing most of the OXPHOS system molecules [78]. Mitochondrial dysfunction is associated with the development of various metabolic diseases. Multiple dysfunctions of the OXPHOS system are associated with clinical manifestations ranging from single lesions in ophthalmological diseases to more extensive lesions or complex multisystem diseases [79,80,81,82]. Decreased OXPHOS activity was found to be associated with the deficiency of NADH:ubiquinone oxidoreductase—Complex I (CI), the first and largest complex of the OXPHOS system [83].

IR, a condition in which body cells present a reduced response to insulin which may cause higher levels of glucose in the bloodstream, as insulin is less effective at helping cells absorb glucose from the blood [84]. This disorder may be a precursor of type 2 diabetes, and it is also associated with a number of other health conditions, including obesity, high blood pressure, and heart disease [84]. It is also a common feature of PCOS [85].

Another important player to discuss is hyperandrogenism which refers to clinical signs and symptoms associated with excessive androgen production or increased sensitivity to androgens in women [59]. The signs include hirsutism, acne, and androgenic alopecia. Androgens, including testosterone and DHEA, are normally present in both men and women but at different levels [86]. In women, androgens play an important role in various physiological processes, such as sexual development, libido, and maintaining bone and muscle mass. In cases of hyperandrogenism, the excessive production of or increased sensitivity to androgens may lead to a range of symptoms and conditions [86]. This clinical syndrome is characterized by the presence of signs and symptoms associated with androgen excess, which may or may not be accompanied by elevated levels of androgens in the blood [59]. Hyperandrogenism is also a hallmark feature of PCOS [87]. Conversely, hyperandrogenemia specifically refers to elevated levels of androgens in the blood [88]. It is noteworthy that hyperandrogenism and hyperandrogenemia are closely related but distinct concepts. Hyperandrogenemia refers to the presence of elevated levels of androgens in the bloodstream, as detected through blood tests [89]. In some cases, hyperandrogenemia and hyperandrogenism occur together, where both elevated androgen levels and associated clinical symptoms are present. This is often seen in PCOS cases manifesting both increased androgen production and hyperandrogenic symptoms [89].

IR and hyperandrogenism are syndromes which are closely related to mitochondrial dysfunction [90]. Insulin is the primary regulator of OXPHOS, and its secretion may directly affect mitochondrial function [91]. This reaction is observed in the opposite direction, as the reduced effectiveness of insulin is a result of increased amounts of ROS. These molecules may induce the abnormal activation of serine/threonine kinase signaling pathways, including c-Jun N-terminal kinases (JNKs), NF-κB or mitogen-activated protein kinases (MAPKs), and affect the phosphorylation of insulin receptors [19]. Studies demonstrated that genes regulating the electron transport chain system in mitochondria were abnormally expressed in PCOS patients [92]. It was also determined that, of all muscle cell organelles, the mitochondria of PCOS patients were the most downregulated cellular component, and mitochondrial electron transport/ubiquinol to cytochrome c was the most downregulated biological process [92]. Abnormal CI function was also shown to be associated with the shape of mitochondrial structures and increased levels of ROS [83], resulting in mitochondrial fragmentation where cellular antioxidant defense systems were not properly balanced.

Studies in the mouse model of obesity revealed that CI, being a mitochondrial acyl carrier protein, acted as an enhancer of mitochondrial metabolism and protected against the development of obesity and IR [93]. Metformin is a widely accepted medication for treating metabolic consequences in women with PCOS [94]. However, the mechanism of action remains unclear, but it is known that metformin has insulin-sensitizing and androgen-lowering properties. AMP-activated protein kinase (AMPK) signaling and mitochondrial respiratory chain CI were suggested as two potential targets for metformin to regulate steroid and glucose metabolism. In addition, one study showed that metformin appeared to reduce oxygen flow through substrates for CI in particular [95]. Thus, it was suggested that the androgen-lowering effect of metformin was mediated by the specific inhibition of CI of the mitochondrial respiratory chain. The direct inhibition of CI by other specific agents should result in reduced androgen production [95] (Figure 1C).

Mitochondrial uncoupling protein 1 (UCP1) is a key component of heat production in brown adipose tissue (BAT) [96]. BAT, also known as brown fat, is a specialized type of the adipose tissue that plays a unique role in thermogenesis and energy expenditure. Unlike white adipose tissue (WAT), which primarily stores energy in the form of triglycerides, BAT is highly metabolically active and generates heat through the process called non-shivering thermogenesis [97]. This mechanism of heat production was observed in women with PCOS exhibiting reduced BAT function resulting from high androgen levels [98]. Subsequently, observations in rats with DHEA-induced PCOS showed a reduced expression of thermogenic genes, UCP1 and the mitochondrial expression of OXPHOS protein, which contributed to developing defects in energy metabolism and BAT activity [99] (Figure 1D).

PCOS might also result from insulin hypersensitivity, which is negatively regulated by UCP2. A significant correlation was demonstrated between increased fasting blood glucose and insulin levels and UCP-2 and CYP11A1mRNA protein levels in women with PCOS [100]. In addition, UCP2 expression in early follicles was significantly higher in PCOS patients than in the controls. This suggests that UCP2 might be involved in the pathophysiology of PCOS and could potentially contribute to the development of the condition. However, more research is needed to fully elucidate the underlying mechanisms and establish a causal relationship.

## 6. Antioxidants as Potential Therapy Methods

Antioxidants were studied for their potential benefits in managing PCOS by reducing OS and improving the symptoms [71,101]. As potent antioxidants, vitamins C and E help neutralize ROS by protecting cells from oxidative damage [102,103]. While their specific role in PCOS is still being explored, vitamins C and E have been investigated for their potential benefits in managing PCOS symptoms and improving the overall health. Some studies suggested that both vitamin C and E supplementation might improve insulin sensitivity and glucose metabolism in women with metabolic syndrome [104]. Vitamins C and E are also involved in the synthesis and metabolism of reproductive hormones [105]. Adequate levels of vitamins may support healthy ovarian function and hormonal balance. Abnormal lipid profiles, including increased levels of triglycerides and cholesterol, are often observed in PCOS. Some studies indicated that vitamin C and E supplementation might have beneficial effects on lipid profiles, potentially lowering triglyceride levels and improving lipid metabolism [105,106,107].

N-acetylcysteine (NAC), a precursor of the antioxidant glutathione is a natural compound which plays a crucial role in cellular antioxidant defense mechanisms and may be helpful in managing the symptoms of PCOS [108]. Potential benefits of NAC supplementation were related to IR reduction, ovulation improvement, and OS reduction in women with PCOS [109,110,111]. NAC may also be beneficial in reducing androgen levels, which can alleviate symptoms such as hirsutism and acne [112]. Additionally, NAC supplementation might help improve menstrual regularity and increase ovulation in women with PCOS, and improve fertility [113,114].

In recent meta-analysis encompassed 18 studies involving a total of 2185 participants was found that NAC supplementation may be beneficial in reducing total testosterone levels and increasing FSH levels in women with PCOS [114]. After accounting for publication bias, NAC was associated with increased estrogen levels. This could have a positive impact on the reproductive system. However, NAC supplementation did not significantly affect other parameters such as the number of ovarian follicles, endometrial thickness, progesterone levels, serum LH levels, or SHBG [114]. Recent studies have shown that oral NAC may be effective in reducing the severity of endometriosis-related pain, including dysmenorrhea, dyspareunia, and chronic pelvic pain. NAC supplementation results in a noticeable depletion in the size of endometriomas and a significant decrease in CA125 serum levels, which suggests a reduction in inflammation and tissue damage. NAC may have a positive impact on fertility among women with endometriosis. A significant number of patients who desired pregnancy were able to achieve it within six months of starting NAC therapy [115].

Inositol is a naturally occurring carbohydrate compound that belongs to the vitamin B family and has gained attention for its potential role in various health conditions, including PCOS [116], especially due to its effects on insulin sensitivity and hormone regulation [117]. Reduced tissue sensitivity to insulin affects 80% of obese and 30–40% of lean women with PCOS [118]. Through the insulin receptor and the receptor for insulin-like growth factor, hyperinsulinemia stimulates androgen production in ovarian theca cells, leading to premature follicular atresia, lack of ovulation and decreased production of SHBG [119,120].

Two main forms of inositol have been studied in this context, myo-inositol (MYO) and D-chiro-inositol (DCI). Inositol may enhance insulin sensitivity, which can be beneficial for individuals with PCOS who often experience IR [121]. Improved insulin sensitivity may lead to better blood sugar control and decreased risk of type 2 diabetes. Inositol supplementation has been associated with improved hormone regulation, including reduction in androgens level [122]. This can help alleviate some of the hormonal imbalances seen in PCOS. Inositol supplementation, particularly MYO, has been studied for its potential to promote regular ovulation in individuals with PCOS [123]. Some studies have suggested that inositol supplementation can lead to a reduction in symptoms commonly associated with PCOS, such as irregular periods, hirsutism, and acne [123]. Inositol may also support weight management efforts in individuals with PCOS, as it can help regulate appetite and metabolism [123]. Within the oocyte, the MYO is predominantly found (99%), with only a small percentage of DCI. Imbalance between inositol forms within the oocyte leads to changes in insulin and FSH secretion [124]. Since both insulin and FSH disturbances are observed in PCOS patients, it seems advisable to compensate for abnormal levels of MYO and DCI and their supplementation in the treatment of both IR and androgenization. Reduction in hyperinsulinemia through inositol therapy leads to an increase in ovulation frequency, restoration of menstrual regularity, reduction in hyperandrogenism and improvement of carbohydrate metabolism, eliminating the vicious circle in PCOS [124]. The IR issue in PCOS patients, abnormal cycles and hormone concentrations, can be effectively controlled and treated by inositol therapies. The use of MYO and DCI combination therapy at the recommended doses affects ovarian inositol concentrations. Furthermore, it seems to be safe for patients, with no side effects observed when used in recommended doses.

However, with regard to the latest data from International evidence-based guideline for the assessment and management of polycystic ovary syndrome, 2023 [125], previous recommendations for inositol supplementation have been refined. The researched areas of inositol application concerned the basic issues related to the management of infertile traits as well as the assessment and treatment of infertility. Meta-analysis performed by an expert group initially identified 43 studies for inclusion in analysis. However, after a thorough integrity check, 14 of these studies were excluded and 29 randomized controlled trials were incorporated into systematic review. Of these, 19 were included in subsequent meta-analysis. When assessing the quality of these studies, it became apparent that ten of them carried a high risk of bias, while 16 were deemed to have either low or moderate risk [125]. Additionally, three studies had an unclear risk of bias. The interventions and comparators studied across these trials exhibited a significant degree of heterogeneity. While evaluating the available evidence, it was determined that, at present, there are insufficient data to make a clear evidence-based recommendation regarding the effectiveness of inositol in all its forms for clinical outcomes. However, there is moderate certainty in the evidence supporting potential benefits of inositol concerning menstrual cycle regulation. On the other hand, the available data supporting its metabolic and hormonal benefits are limited, and the certainty of this evidence is very low [125].

According to *Pharmacological treatment for non-fertility indications*, inositol supplementation was included in the Second-line Pharmacological Therapies section of the latest 2023 guidelines [125]. As the algorithm implies, in women with PCOS, the use of inositol in any of its forms can be considered based on individual preferences and values, given its relatively low risk profile and potential benefits in addressing specific biochemical aspects of PCOS. However, it is important to note that the clinical benefits of inositol for promoting ovulation, reducing hirsutism, or aiding in weight management are limited. Furthermore, women should be advised to inform their healthcare providers if they decide to use inositol preparations. These recommendations represent an evolution from the 2018 Guideline [126] and emphasize the importance of shared-informed decision making regarding the use of products that may offer potential benefits but lack robust evidence of harm.

While metformin primary mechanism of action is to reduce glucose production in the liver and improve insulin sensitivity in peripheral tissues, it also appears to have some potential antioxidant effects [127]. Metformin’s primary action involves altering cellular energy metabolism by inhibiting the mitochondria’s ability to produce energy through certain pathways [128]. This alteration can lead to a reduction in the production of ROS within the mitochondria, which are a major source of ROS in the cell. Metformin activates an AMPK enzyme, which plays a key role in regulating cellular energy balance. AMPK activation can lead to various cellular effects, including increased antioxidant defenses [129].

Metformin has been shown to have anti-inflammatory effects by affecting certain inflammatory pathways. Inflammation is closely linked to OS, and reducing inflammation can indirectly contribute to antioxidant effects [130]. Endothelial dysfunction, a common feature in diabetes is associated with OS. Metformin has been shown to have protective effects on endothelial cells, possibly due to its antioxidant properties [131]. Furthermore, metformin influences hormone levels in individuals with PCOS, particularly by reducing the levels of androgens. Since hormonal imbalance is a hallmark of PCOS, metformin’s ability to restore more balanced hormone levels could help alleviate some of the symptoms associated with the condition [132].

Selenium is a component of antioxidant enzymes such as glutathione peroxidase, which help neutralize ROS and reduce OS. Given that selenium’s antioxidant properties could be beneficial in reducing oxidative damage. Selenium may play a role in improving insulin sensitivity, but not direct hormonal effect [133]. Additionally, selenium is known to modulate immune responses, inflammation, and impact hormonal balance. Chronic low-grade inflammation is associated with PCOS, and selenium’s anti-inflammatory effects might contribute to better management of PCOS symptoms [134,135]. Selenium is a crucial component of enzymes involved in the production and conversion of thyroid hormones [136]. Thyroid peroxidase (TPO) enzyme is essential for the production of thyroid hormones. TPO catalyzes the iodination of tyrosine residues in thyroglobulin, a protein produced by the thyroid gland, which is a key step in the synthesis of thyroid hormones, the triiodothyronine (T3) and thyroxine (T4). Without adequate selenium, TPO activity can be compromised, leading to thyroid hormone imbalances [137]. In turn, deiodinase enzymes are responsible for converting one form of thyroid hormone (T4) into the more active form (T3) and regulating their availability in various tissues [138]. Selenium is required for the proper function of these deiodinase enzymes [139]. Some individuals with PCOS may also experience thyroid dysfunction [140,141]. Hormonal imbalances and IR associated with PCOS can potentially impact thyroid function. Thyroid dysfunction, particularly conditions such as hypothyroidism, can contribute to various PCOS symptoms, including irregular menstrual cycles and weight management challenges [140].

Importantly, although the use of these antioxidants seems promising, the optimal dosage, duration, and effectiveness in PCOS management are still subject to ongoing research. Furthermore, individual responses to antioxidants may vary, and it is recommended to consult a healthcare professional before initiating any antioxidant supplementation. In addition to antioxidant supplementation, incorporating a balanced diet rich in fruits, vegetables, and whole grains may provide a natural source of antioxidants. Lifestyle modifications, such as regular exercise and weight management, can also contribute to reducing OS in PCOS.

An important, separate issue of antioxidants usage is their synergistic action which refers to the enhanced effects that occur when multiple antioxidants are used together, often resulting in greater overall benefits compared to using individual antioxidants alone. Antioxidants operate through various mechanisms. Some antioxidants might neutralize specific types of ROS, while the others repair oxidative damage or enhance the body’s natural antioxidant defenses. By combining antioxidants with diverse mechanisms, a broader range of oxidative stress-related issues could be potentially cover. Vitamins D and E, coenzyme Q10 (CoQ10) and inositols are among the antioxidants that have been investigated for their potential benefits in managing PCOS-related symptoms, particularly in relation to IR, inflammation, and OS [142]. Meta-analysis study showed that inositols could be advantageous in increasing SHBG and improving glycolipid metabolism. In turn, vitamin E might be beneficial for reducing total testosterone levels and increasing SHBG. CoQ10, whether used alone or combined with vitamin E, could be useful in decreasing IR as measured by homeostatic model assessment of IR (HOMA-IR) [142]. Co-supplementation with certain combinations has been associated with improvements in inflammatory status and antioxidant capacity. The combination of probiotics and selenium or vitamin E with omega-3 fatty acids could provide benefits in reducing inflammation and enhancing antioxidant capacity. Inositol enrichment with vitamins B group could have positive effects on factors such as inflammation, antioxidant capacity, and overall health [143].

## 7. Conclusions

While the association between OS and PCOS is increasingly recognized, further research is needed to fully elucidate the role of OS in PCOS pathophysiology. Investigating the mitochondrial function, ATP production, and OXPHOS efficiency in PCOS may provide further insights into the metabolic and hormonal dysregulation associated with the condition. Various antioxidants, such as NAC, vitamin E, and alpha-lipoic acid, have been proposed as potential therapeutic agents to reduce OS and improve metabolic and reproductive outcomes in women with PCOS. Lifestyle interventions, including exercise and dietary modifications, have also been shown to reduce OS and improve insulin sensitivity and fertility in PCOS patients. Understanding the links between OS and PCOS is crucial for developing effective treatment strategies to improve the quality of life of women affected by this condition.

## Figures and Tables

**Figure 1 ijms-24-14126-f001:**
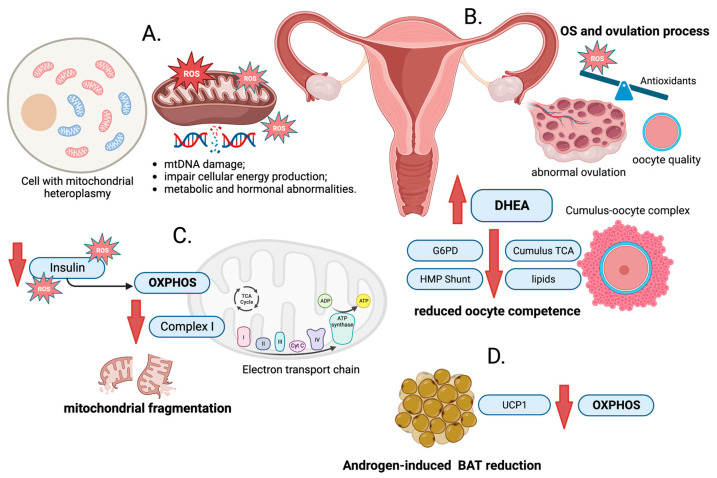
Schematic presentation of the associations between oxidative stress (OS) and polycystic ovary syndrome (PCOS). (**A**). Mitochondria are the energy-producing structures in cells. The close proximity of mitochondrial DNA (mtDNA) to the source of reactive oxygen species (ROS) production and the limited protective mechanisms in mitochondria contribute to its susceptibility to oxidative damage and can result in various types of mutations. (**B**). Ovulation is a crucial event in the menstrual cycle, where a mature egg is released from the ovary, ready for potential fertilization. In small amounts, ROS are essential for normal physiological processes; however, excessive production of ROS, meaning oxidative stress (OS), can be harmful to cells and tissues. The reduction in the rate of ovulation and prevention of cumulus expansion observed after antioxidant administration might be due to the antioxidants’ neutralizing effects on ROS and disruption of the delicate balance required for successful ovulation and cumulus expansion. The citric acid (TCA) cycle is a fundamental metabolic pathway that generates energy. Citrate formation is crucial for oocyte competence process. In the oocytes of DHEA-induced PCOS mice, TCA, glucose-6-phosphate dehydrogenase (G6PD) activity, and lipid content is decreased, suggesting abnormal metabolism in the TCA cycle and the pentose phosphate pathway (HMP Shunt), which could negatively impact oocyte function. (**C**). Decreased oxidative phosphorylation (OXPHOS) activity is associated with the deficiency of NADH: ubiquinone oxidoreductase (Complex I, CI). Insulin plays a crucial role in regulating OXPHOS activity. It affects mitochondrial function by influencing the electron transport chain and ATP production. Increased ROS production can have a negative impact on insulin sensitivity, leading to insulin resistance. (**D**). Mitochondrial uncoupling protein 1 (UCP1) allows protons to re-enter the mitochondrial matrix, uncoupling the OXPHOS process from ATP production and releasing energy as heat (non-shivering thermogenesis). In women with PCOS, reduced brown adipose tissue (BAT) function was observed, which may be attributed to high androgen levels.

## Data Availability

Not applicable.

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
