# Peer review of "Polycystic Ovary Syndrome and Oxidative Stress—From Bench to Bedside"

_ijms, 2023, doi:10.3390/ijms241814126_

Round 1

Reviewer 1 Report

This article discusses the relationship between OS and PCOS from three aspects detailedly: ROS production and mitochondrial DNA damage, oxidative stress, oxidative phosphorylation and antioxidants as potential therapy methods and OS has been linked to the key features of PCOS, such as insulin resistance, hyperandrogenism, and ovulation disorder from this article. The overall logical structure of the article is complete, from basic research to clinical treatment, but to address some issues, I would like to make some suggestions in this review:

1. In the first point, only mitochondrial DNA damage is described in detail, not ROS production, what is the specific mechanism of ROS production in PCOS?

2. In the third point, the article focuses on OS and ovulation of PCOS, but the description of other characteristic aspects of PCOS is too brief. For example, hyperandrogenism is an important characteristic of PCOS, and the research on this aspect is relatively rich at present, so the description in this paper is not comprehensive.

3. There also seems to be a lack of description of hyperandrogenism in the high summary of abstracts.

4. At present, there have been several reviews on OS in PCOS and what are the advantages and innovation points of this article?

Author Response

Dear Reviewer,

We do appreciate valuable Reviewer’ comments, which we carefully considered. We made all the required changes to the revised manuscript and believe that thanks to them our article gained value and now can be considered for publication in IJMS.

All changes that were made are clearly marked in blue throughout the resubmitted manuscript.

Responses to the comments:

We agree with the comments that our manuscript should be enriched with the paragraphs provided by the Reviewer. Accordingly, we have made changes to the manuscript. In details:

  1. In the first point, only mitochondrial DNA damage is described in detail, not ROS production, what is the specific mechanism of ROS production in PCOS?

In the revised manuscript, Section 2. ROS production and mitochondrial DNA damage has been divided to Section 2. ROS production and Section 3. Mitochondrial DNA damage, and a paragraph regarding mechanisms of ROS production in PCOS has been added (lines: 78-133). In addition, in section 4. Oxidative stress and PCOS, relevant sentences were included (lines: 224-234).

  1. In the third point, the article focuses on OS and ovulation of PCOS, but the description of other characteristic aspects of PCOS is too brief. For example, hyperandrogenism is an important characteristic of PCOS, and the research on this aspect is relatively rich at present, so the description in this paper is not comprehensive.

We agree with the Reviewer that we have not covered the relationship of OS and hyperandrogenism in our manuscript. Therefore, section 4. Oxidative stress and PCOS has been expanded by adding a paragraph concerning OS and hyperandrogenism (lines: 250-297).

  1. There also seems to be a lack of description of hyperandrogenism in the high summary of abstracts.

In the revised manuscript, we also included a modified version of the abstract.

  1. At present, there have been several reviews on OS in PCOS and what are the advantages and innovation points of this article?

We believe that our manuscript provide an in-depth examination of the connections between OS and PCOS, touching upon various aspects of the condition, including metabolic and hormonal dysregulation, ovarian follicles, fertility, and insulin resistance. This comprehensive approach can be advantageous for readers seeking a thorough understanding of the topic. Our article highlights the potential impact of mitochondrial mutations on OS and PCOS. This is an innovative aspect as it delves into the molecular mechanisms that may contribute to the condition, specifically impaired oxidative phosphorylation (OXPHOS) and reduced ATP production. Manuscript emphasizes the relationship between OS and fertility by discussing the detrimental effects of ROS on oocytes and granulosa cells within ovarian follicles. This link between OS and fertility is particularly important, as fertility issues are a significant concern for many women with PCOS. The mention of antioxidants as potential therapies for PCOS is a practical aspect of the article. Discussing potential treatment strategies derived from the understanding of OS in PCOS can be beneficial for clinicians and researchers looking for avenues to improve the quality of life for affected individuals.

We believe that this article, while drawing on existing literature, also provide new insights, which could be interesting and inspiring for researches seeking the latest developments in the field.

Reviewer 2 Report

This is an interesting review of the links between oxidative stress and PCOS. While the paper's objective is important, it does have a few flaws that should be addressed.

The captions in Figure 1 are excessively long and difficult to comprehend. I suggest rewriting them to provide concise summaries of the key messages.

In Section 5, titled "Antioxidants as Potential Therapy Methods," I recommend referencing the latest ESHRE Guide: "International evidence-based guideline for the assessment and management of polycystic ovary syndrome 2018" available at https://www.eshre.eu/Guidelines-and-Legal/Guidelines/Polycystic-Ovary-Syndrome. In this regards, it would be valuable to incorporate the most recent evidence regarding the role of N-Acetyl Cysteine and Inositol in TRA results. Highlighting the recommendations on whether these therapies should currently be considered experimental therapy in PCOS.

Author Response

Dear Reviewer,

We do appreciate valuable Reviewer’ comments, which we carefully considered. We made all the required changes to the revised manuscript and believe that thanks to them our article gained value and now can be considered for publication in IJMS.

All changes that were made are clearly marked in blue throughout the resubmitted manuscript.

Responses to the comments:

  1. The captions in Figure 1 are excessively long and difficult to comprehend. I suggest rewriting them to provide concise summaries of the key messages.

As was suggested, we have shortened the caption for Figure 1.

  1. In Section 5, titled "Antioxidants as Potential Therapy Methods," I recommend referencing the latest ESHRE Guide: "International evidence-based guideline for the assessment and management of polycystic ovary syndrome 2018" available at https://www.eshre.eu/Guidelines-and-Legal/Guidelines/Polycystic-Ovary-Syndrome. In this regards, it would be valuable to incorporate the most recent evidence regarding the role of N-Acetyl Cysteine and Inositol in TRA results. Highlighting the recommendations on whether these therapies should currently be considered experimental therapy in PCOS.

As was recommended, in section . Antioxidants as potential therapy methods, we referred to the latest recommendations according to the ESHRE Guide regarding inositol supplementation. The relevant paragraph has been added to the section (lines: 457-487).

However, currently, the latest ESHRE recommendations do not include data on NAC supplementation. Nevertheless, these data is being collected successively. In the revised version of the manuscript, the paragraph on NAC supplementation has been extended (lines: 411-424).

Round 2

Reviewer 1 Report

The revision is satisfied.  There is no more comments.